# Beyond the Digital Competencies of Medical Students: Concerns over Integrating Data Science Basics into the Medical Curriculum

**DOI:** 10.3390/ijerph192315958

**Published:** 2022-11-30

**Authors:** Diana Lungeanu, Alina Petrica, Raluca Lupusoru, Adina Maria Marza, Ovidiu Alexandru Mederle, Bogdan Timar

**Affiliations:** 1Center for Modeling Biological Systems and Data Analysis, “Victor Babes” University of Medicine and Pharmacy, 300041 Timisoara, Romania; 2Department of Functional Sciences, Faculty of Medicine, “Victor Babes” University of Medicine and Pharmacy, 300041 Timisoara, Romania; 3Department of Surgery, Faculty of Medicine, “Victor Babes” University of Medicine and Pharmacy, 300041 Timisoara, Romania; 4“Pius Brinzeu” Emergency County Clinical Hospital, 300723 Timisoara, Romania; 5Multidisciplinary Center for Research, Evaluation, Diagnosis and Therapies in Oral Medicine, “Victor Babes” University of Medicine and Pharmacy, 300041 Timisoara, Romania; 6Emergency Municipal Clinical Hospital, 300079 Timisoara, Romania; 7Center for Molecular Research in Nephrology and Vascular Disease, “Victor Babes” University of Medicine and Pharmacy, 300041 Timisoara, Romania; 8Second Department of Internal Medicine, Faculty of Medicine, “Victor Babes” University of Medicine and Pharmacy, 300041 Timisoara, Romania

**Keywords:** medical education, biomedical informatics, information and communications technology, biostatistics, contextual learning

## Abstract

Introduction. Data science is becoming increasingly prominent in the medical profession, in the face of the COVID-19 pandemic, presenting additional challenges and opportunities for medical education. We retrospectively appraised the existing biomedical informatics (BMI) and biostatistics courses taught to students enrolled in a six-year medical program. Methods. An anonymous cross-sectional survey was conducted among 121 students in their fourth year, with regard to the courses they previously attended, in contrast with the ongoing emergency medicine (EM) course during the first semester of the academic year 2020–2021, when all activities went online. The questionnaire included opinion items about courses and self-assessed knowledge, and questions probing into the respondents’ familiarity with the basics of data science. Results. Appreciation of the EM course was high, with a median (IQR) score of 9 (7–10) on a scale from 1 to 10. The overall scores for the BMI and biostatistics were 7 (5–9) and 8 (5–9), respectively. These latter scores were strongly correlated (Spearman correlation coefficient R = 0.869, *p* < 0.001). We found no correlation between measured and self-assessed knowledge of data science (R = 0.107, *p* = 0.246), but the latter was fairly and significantly correlated with the perceived usefulness of the courses. Conclusions. The keystone of this different perception of EM versus data science was the courses’ apparent value to the medical profession. The following conclusions could be drawn: (a) objective assessments of residual knowledge of the basics of data science do not necessarily correlate with the students’ subjective appraisal and opinion of the field or courses; (b) medical students need to see the explicit connection between interdisciplinary or complementary courses and the medical profession; and (c) courses on information technology and data science would better suit a distributed approach across the medical curriculum.

## 1. Introduction

Information and communications technology (ICT) has pervasively influenced our lives for years, with the COVID-19 pandemic precipitating great changes in this area across society. Education is undergoing a swift and dramatic transformation in paradigm, in the context of digital policies aiming beyond mere digital literacy, such as those of the European Union [1]. Medical educators face the challenges of re-thinking their teaching approaches and adapting their courseware to synchronous and asynchronous educational activities, or even restructuring the whole curriculum [2,3,4,5,6]. They are compelled to find solutions to incentivize their students’ resilience and help them develop higher-order thinking. Communities of educators have teamed up to develop and share workable solutions under Creative Commons licenses or as open-source software implementations [2,6]. In recent years, data science has emerged as an interdisciplinary field encompassing informatics, statistics, computer science, data management and mathematics, borrowing tools from machine learning and data mining. It has been instrumental in societal transformation, including education (where revolutionizing learning analytics makes use of the stakeholders’ data sense-making abilities) and the medical profession (where insights from data are of crucial importance). To develop their medical expertise, medical students and residents start by attending formal education programs and then journal papers serve to continue this education for the majority of medical professionals. Knowledge of data science and an ability to understand the results of research papers can be decisive complements to clinical practice [7,8] and medical students should become aware of that potential [9]. Moreover, there are controversial pros and cons regarding the employment of learning analytics in education [10,11] and the appropriate integration of data science into medical programs [12,13,14,15].

Worldwide, medical curricula have included interdisciplinary courses on ICT and biostatistics. International boards of experts have issued recommendations and guidelines for such courses and specializations at different levels [16,17,18,19,20]. At our University, since 1992, the medical curriculum has included mandatory courses on medical informatics and biostatistics and we have been trying to align the courses’ syllabi to international standards [21]. Four years ago, we re-designed the courses as a coordinated introduction to the basics of data science in the first and second year of the academic medical program, aiming to yield results after one six-year academic cycle. Nevertheless, the transformational context of the pandemic has called for reflection and a rethinking of priorities, so we face decisions about optimizing the transfer of the gained know-how into new courseware and the associated educational approaches we should deploy.

Taking all of these factors into consideration, we conducted a retrospective appraisal of the two existing biomedical informatics (BMI) and biostatistics courses, as a survey of the fourth-year medical students, in the context of the COVID-19 pandemic; this assessment was an exploratory analysis and was run in contrast with the emergency medicine (EM) course, during the clinical stage of the undergraduate medical program. The research objectives were as follows: (a) to assess the residual knowledge of BMI and biostatistics; (b) to collect students’ opinion and attitudes towards the two previous data science courses as opposed to the EM course; and (c) to analyze the possible pandemic-related confounders or determinants of individual perceptions in regard to the quality of the above-mentioned courses, such as students’ overall satisfaction with life, perception of work effectiveness and online professional activity, level of depression, health and support from the university.

## 2. Materials and Methods

### 2.1. Study Design and Participants

An online cross-sectional survey was employed for data collection: an anonymous questionnaire (implemented using Google Forms) was distributed to the students attending the EM course during the first semester of the academic year 2020–2021, when all activities went online; students were requested to distribute the questionnaire to their peers (i.e., same generation, attending a similar clinical program). The survey was active mid-semester, between 27 November 2020 and 6 December 2020 (it closed after three consecutive days with no answer). It started with information about the study’s goals and the measures taken to assure personal data protection. For each individual, actual data collection proceeded after informed consent had been granted (required confirmation was included as the questionnaire’s first item).

Figure 1 shows the study flow diagram. BMI, biostatistics and EM courses are mandatory in the six-year medical program of our university. The survey targeted four areas, implemented as distinct sections of the questionnaire: (a) level of depression, perception of work effectiveness and overall satisfaction with life and online professional activity during the ongoing COVID-19 pandemic; (b) residual knowledge of BMI and biostatistics and opinion of the previous BMI and biostatistics courses; (c) feedback about the ongoing EM course; and (d) opinion of the prospective applicability of e-learning and online instruments in medical education.

Depression was measured with the nine-item Patient Health Questionnaire (PHQ-9), validated for the Romanian population [22,23] and proven to be a valid screening tool for depression in various settings (i.e., not only in clinical settings, as initially developed) and also for the general population [24,25,26,27].

Residual knowledge of BMI and biostatistics was gauged with 14 multiple choice questions. These questions were designed for this particular investigation, based on the courses’ syllabi and on our repository of questions used in ordinary end-of-course examinations. The answers were manually coded by two independent reviewers and any disagreements were resolved by consensus. Appendix A shows the syllabi for the biomedical informatics and biostatistics courses in the academic years 2017–2018 and 2018–2019, respectively.

Students’ opinions and feedback were collected on five-point Likert-type scales. The overall opinion of each course was recorded as integer marks between 1 and 10. For the ongoing EM course, opinions were separately collected for lectures and practical classes, for this was the first time the students had experienced online clinical activities. The face validity of the questions designed for this project was assured by the Delphi technique in the development process; no prior formal validation was conducted.

Appendix A details the tools employed for data collection and presents the full questionnaire.

### 2.2. Data Analysis

Descriptive statistics comprised the mean and standard deviation for age, a normally distributed numerical variable. Normality was tested with the Shapiro–Wilk statistical test; subsequently, the ANOVA parametric test was used for the statistical significance of observed differences between the genders. Scale scores were treated as rank variables, described by the median and the inter-quartile range (IQR); non-parametric methods were further employed for statistical analysis, such as the Kruskal–Wallis test, Wilcoxon signed rank test and Spearman coefficient of correlation. Categorical variables were described by the observed frequencies (i.e., counts) and their corresponding percentages; the Chi-square test was applied for statistical significance (either asymptotic, or Monte-Carlo simulation based on 10,000 samples).

For meaningful groups of questions within the anonymous questionnaire (such as PHQ-9, residual knowledge, self-assessed knowledge, or course usefulness), the actual reliability of measurements was assessed based on Cronbach’s alpha. Values over 0.8 were considered as proving good internal consistency.

For the group scales concerning the level of knowledge (either measured or self-assessed), in addition to the raw totals, a zero-max rescaling was applied, with the maximum set at 100%. Additional descriptive statistics were also presented for these rescaled scorings, treated as ranks.

The statistical analysis was conducted at a 95% level of confidence (i.e., 5% level of statistical significance). All reported probability values were two-tailed and highly significant values were also marked. Data were analyzed with the statistical software IBM SPSS v. 20.0 (Armonk, NY, USA) and the statistical packages R v.4.0.5 (https://cran.r-project.org/, accessed on 22 September 2022).

## 3. Results

One hundred and twenty-one students aged between 21 and 27 years responded: 28 were male, 84 female and 9 preferred not to declare their gender. With only one exception (a male student), respondents answered all questions. The EM course during which the questionnaire was distributed enrolled 79 students; thus, high response and dissemination rates could be inferred, exceeding by 50% the number of students with whom EM professors directly interacted during that time.

Table 1 synthesizes the results of the short quiz aimed at assessing the residual knowledge on data science (presumed to have been acquired during the BMI and biostatistics courses attended two years earlier), in parallel with the self-assessed level of knowledge. While Cronbach’s alpha for the self-assessment was high (based on five items), the value for the 14-item quiz was low (although the number of items was almost three times higher). The students’ opinions of the approach in terms of the examination and usefulness of the two courses were fair and the high values of their respective Cronbach’s alpha coefficients suggest the good reliability of these collected data. Table 1 also includes descriptive statistics for the marks, capturing the overall opinion of each of the two previous courses. It is notable that opinions were unanimous across the genders, on all aspects regarding the two previous data science courses.

Descriptive statistics of the self-assessed level of acquired EM knowledge are presented in Table 2, together with students’ opinions in regard to e-learning practical classes and prospective online examination on the EM course. Although the online activities and examinations were seen as poor substitutes for traditional approaches, the respondents acknowledged a high level of newly acquired knowledge concerning EM. This high regard for the EM course was also reflected by the overall marks given to the lectures and practical classes. When asked about their familiarity with and opinion of 360-degree video technology used in teaching EM course (Q53 and Q54), respondents were optimistic about its educational usefulness, although their declared familiarity with this technology was one level below their confidence in its benefits. Similarly to the opinions about previous courses, there was no significant difference across the genders in regard to the EM course and its associated activities.

Table 3 presents associations concerning the BMI and biostatistics courses as a two-by-two correlation matrix between the measured residual knowledge, self-assessed level of knowledge, opinions of the examinations and usefulness of previous courses and overall marks students gave to the courses. There was an insignificant correlation between the residual and self-assessed level of knowledge (Spearman coefficient R = 0.107, *p* = 0.246), but the high level of correlation between the two overall marks given to the BMI and biostatistics courses is noteworthy. At the same time, the residual knowledge did not correlate with any of the students’ subjective appraisals, namely their perceptions or opinions. On the other hand, all of these perceptions and opinions were significantly two-by-two correlated.

Table 4 synthesizes the two-by-two correlation between the overall marks given to each course or particular activity; there was a strong and highly significant correlation between similar activities: (a) the ongoing EM activities (lectures and practical classes); and (b) the previous BMI and biostatistics courses. In contrast, the correlation between the appreciation of EM activities and previous data science courses was rather weak (although it was statistically significant).

Appendix A comprises four additional tables, containing results considered helpful for contextualizing the above data; we regarded them as non-essential for conveying the main message of this paper, but as still important for further exploring the possible confounders in the feedback we collected in a limited time window in the fall of 2020.

Appendix A presents the students’ opinion and concerns regarding their online professional activity, perception of work effectiveness, depression and overall satisfaction with life. Except for life satisfaction, there was no significant difference between the genders. 

Appendix A shows the students’ perception of their own abilities in employing the ICT technology for independent online activities, their motivation towards attending such activities and the prospective applicability of e-learning and online activities in medical education: they seemed confident in their ICT abilities, but their levels of motivation and confidence in the educational validity of these existing curriculum activities were lower. 

Two supplementary correlation matrices (Appendix A) show the strength of association between the levels of satisfaction with education, life, health and perceived support from the university, and the overall marks students gave to the data science and EM courses, respectively. The marks did not appear to depend on the levels of depression or satisfaction.

## 4. Discussion

Students responded to the survey request with enthusiasm and eagerness to provide feedback, although they were not fully satisfied with their health, professional life, or life in general: the results shown in Appendix A help to put the course feedback in a larger context, regarding respondents’ satisfaction and personal perspective on their profession and life. Our survey shows that, as the online educational activities gained momentum and all stakeholders acquired much needed know-how in the new academic context, the students’ professional lives improved. Nevertheless, their level of satisfaction remained rather low in regard to the perceived support from the university and with life in general. Our findings confirm other reported concerns related to education during the COVID-19 pandemic and students’ struggle to adapt [28,29,30,31], or issues regarding the basis of present medical education [32].

We found a weak but statistically significant correlation between the perceptions and attitudes concerning the courses and the levels of satisfaction with regard to life, health and perceived support from the university. However, we found no evidence to suggest that frustration or depression engendered the feedback on any of the surveyed courses.

We contrasted students’ opinions of two different course profiles, at distinct stages of the undergraduate medical program: on the one hand, the two BMI and biostatistics courses, previously taught during the preclinical stage; and on the other hand, the ongoing activities of the EM course, during the clinical stage. Online teaching and learning EM were obviously challenging for teachers and students alike. Nevertheless, students seemed willing to overcome the difficulties and seized every opportunity to improve their preparedness for their chosen career, i.e., the medical profession, in line with other reports about technology-enhanced teaching being well-received by medical students [33]. Notably, compared with the results reported by Baashar et al. [33], the students who participated in our project obtained good marks in further objective EM examinations at the end of that semester, thus confirming their good preparedness, in addition to their commitment and declared motivation [34].

The feedback about the two previous data science courses reveals students’ disconnection from the topic: self-assessed and measured residual knowledge were not correlated. In addition, the reliability of the measured knowledge was low, an issue that could have two roots: (a) the questions were irrelevant, i.e., they lacked validity; and (b) there were too few items for such a broad range of subjects and concepts. In contrast, the respondents seemed rather confident in their knowledge and practical abilities in regard to the actual employment of ICT in their current professional activities. Moreover, the self-assessed knowledge significantly correlated with the opinions of the courses and their usefulness. When the two courses were re-designed in 2017, they were intended as a coordinated introduction to biomedical data science and it seemed we achieved that goal: the participants’ perceptions and overall feedback on the courses strongly correlated. Nonetheless, the feedback itself was reserved: the courses did not appear to fit into the undergraduate medical program and two different underlying reasons might explain this feedback we received. Firstly, having data science courses in the preclinical stage might be too early; biostatistics might be better suited to the later stages of medical education, when the need for making sense of data is more apparent. Secondly, such courses conveying interdisciplinary or complementary knowledge might seem non-essential to the medical profession and therefore not a priority for students. On the other hand, biomedical informatics in the early stages would help to make sure that the necessary digital competencies are mastered. We also acknowledge the strong correlation between the marks given to the two courses, which suggests that the students did not perceive them as disconnected. To sum up, having multiple course modules along the medical curriculum in a manner of progressive complexity might be more effective. Thus, questions and concerns arise in regard to the priorities of the basic curriculum. Personal career preferences towards immediate action or research activities might also play a role in prioritizing learning efforts, as discussed by Hammaker et al. [35].

Apparent usefulness in the medical profession seems to be the keystone of students’ involvement and commitment: the specific problems to which data science would bring solutions should be perceptible to the medical students. Courses that could be seen as new to the profession, such as informatics and biostatistics, should blend into the curriculum, with evident applicability. Established authors have advocated the importance of data science in medical education and timely implementation approaches have been suggested [9,36,37]. The United States’ National Library of Medicine offers data science and informatics training programs designed for researchers and healthcare workers [38]. The disconnection we found between self-assessed and measured knowledge might have been generated by our failure in conveying the weighty concepts of biomedical data science. On the other hand, the pattern that emerged from the feedback we collected was the students’ self-confidence in their practical abilities to use information technology for their everyday needs, combined with no perceived necessity for additional insight into medical data. Making sense of data did not constitute a concern for the students we surveyed, so this should be a concern for educators.

Innovative educational approaches, such as active learning, self-directed and implicit learning, flipped classrooms, or gaming, might help us put the basics of data science into the right context, as they have already been brought into the discussion [2,5,6,39,40,41,42]. In 2013, Bok et al. [43] proposed workplace learning in medical education as a context for measuring and documenting competence development, together with formative feedback in addition to summative decisions. Contextual learning might be a fruitful implementation of such courses (i.e., courses with no foreseen or immediate usefulness for beginners). Moreover, acquaintance with the basics of data science would bring about a two-fold gain for medical students: (a) helping them to become better future professionals; and (b) helping them to understand the concept of learning analytics and become actively involved in self-regulating their own learning, as has been suggested for years [9]. Based on the experience from other areas of higher education, medical educators can help their students to gain meta-cognitive understanding and build self-regulating abilities [44,45].

In a broader context, apart from academics and medical professionals, other stakeholders should also be involved as agents of this change in medical education, such as healthcare organizations and medical students [3,46,47]. However, finding the balance between learners’ personal preferences and an adherence to academic requirements and standards is a challenge that might generate frustration for some medical students. Although we acknowledge that an effective education is a collective endeavor, we believe it remains the duty of the faculty and professors to find functional approaches in the pursuit of equilibrium. Moreover, to this end, educational approaches should be tailored to each curricular theme.

### Generalization Limitations

The main limitations were generated by the cross-sectional design of the survey and the limited number of questions in each of its sections. The low reliability of the residual knowledge assessment was a caveat we acknowledge throughout the paper.

The opinions were collected from students of one university and one generation, thus representing a limited sample of feedback information to draw conclusions from. An additional limitation was produced by the opportunistic nature of this investigation during the COVID-19 pandemic and the skipping of the formal validation stage during questionnaire deployment. 

With these limitations in mind, our exploratory investigation identifies opinions and issues for further discussion and investigation on a larger scale and using intervention-based designs, rather than making categorical judgments or offering purportedly miraculous solutions.

On the other hand, our survey has the advantage of data being collected during the very time when the traditional approaches reached their limits and when information technology and data-related sciences became strongly engaged in medical education. This unprecedentedly challenging situation has created both hurdles and opportunities in medical education and our efforts were focused on measures to not miss the latter.

## 5. Conclusions

We believe that our survey contributes evidence on the need to provide an adequate medical background and rationale for all complementary courses in the medical curriculum and for biomedical data science in particular. The results also provide additional evidence to support the broadening of educational approaches themselves and the incorporation of contextual or implicit learning, as well as active or self-directed learning, into the interdisciplinary courses of the medical curriculum.

Our message, asking for integration and contextualization, is aimed at medical education stakeholders at various levels, from institutional policy makers to curriculum designers and teachers themselves. We opt for a distributed approach, which offers a beneficial solution in terms of contextualizing data science assets for the medical profession and fostering long-term practices in terms of the use of data in daily medical practice.

## Figures and Tables

**Figure 1 ijerph-19-15958-f001:**
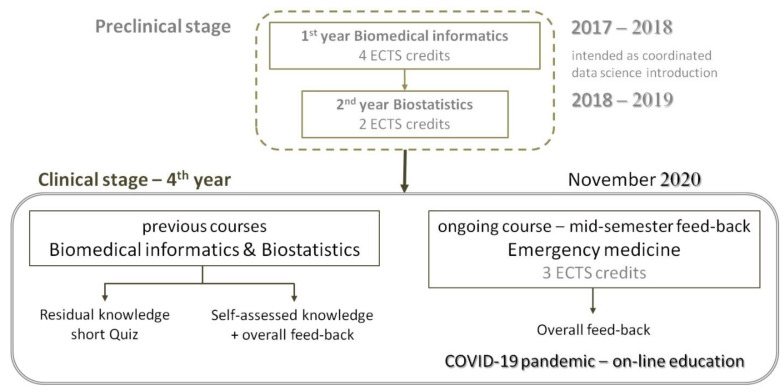
Study flow diagram.

**Table 1 ijerph-19-15958-t001:** Measured residual and self-assessed knowledge of BMI and biostatistics and students’ opinions of the two courses (including examinations, perceived usefulness and overall appraisal for each).

Question(s)/Variable		All	Males	Females	Not Declared	*p*-Value ^(a),(b)^
		N = 121	N = 28	N = 84	N = 9	
Q21 to Q34	Residual knowledge Total28 ^(a),#^	12 (10–15)	12 (10–15) ^#^	12 (10–15)	10 (8–14)	0.608
	14 items, Cronbach’s alpha = 0.369					
	Percentage scale residual knowledge ^(a),#^	57.14 (47.62–71.43) ^#^			47.62 (38.10–66.67)	–
Q35 to Q39	I learnt Total25 ^(a)^ Biomedical data science	16 (14–19)	16 (15–20.5)	16 (14–19)	13 (12–21)	0.58
	5 items, Cronbach’s alpha = 0.929					
	Percentage scale I learnt ^(a)^	64 (56–76)	64 (60–82)	64 (56–76)	52 (48–84)	–
Q40	Examination Total10 ^(a)^ Biomedical data science	8 (6–9)	8 (7–9)	8 (6–9)	6 (6–8)	0.123
Q41	2 items, Cronbach’s alpha = 0.868					
Q42	Usefulness Total10 ^(a)^ Biomedical data science	8 (6–9)	8 (7–9)	8 (6–9)	6 (6–8)	0.798
Q43	2 items, Cronbach’s alpha = 0.826					
Q44	Overall opinion BMI course ^(b)^	7 (5–9)	6.5 (5–8)	8 (5–9)	7 (3–9)	0.449
Q45	Overall opinion of the biostatistics course ^(b)^	8 (5–9)	8 (6–9)	8 (5–9)	8 (4–9)	0.967

^(a)^ median (IQR) for the totals (sum of the items); Kruskal–Wallis statistical test for significance of observed differences between the three gender groups; ^(b)^ mark between 1 and 10; median (IQR); Kruskal–Wallis statistical test for significance of observed differences between the three gender groups. Notation: IQR, inter-quartile range; ^#^, one missing value for a male respondent.

**Table 2 ijerph-19-15958-t002:** Opinions of the ongoing EM course, including separate appraisals for lectures, practical activities and 360-degree video scenarios employed as online educational instruments.

Question(s)/Variable		All	Males	Females	Not Declared	*p*-Value ^(a),(b),(c)^
		N = 121	N = 28	N = 84	N = 9	
Q47	I learnt EM Total15 ^(a)^	13 (9–15)	12 (9–15)	15 (9–15)	14 (12–15)	0.189
	3 items, Cronbach’s alpha = 0.929					
	Percentage scale I learnt EM ^(a)^	86.67 (60–100)	80 (60–100)	100 (60–100)	93.33 (80–100)	–
Q48	E-learning is a valid substitute for clinical practice ^(b)^	2 (1–3)	2 (1–3)	2 (1–3)	2 (1–3)	0.704
Q49	Online examination is valid ^(b)^	3 (1–3)	2 (1–3)	3 (1–3)	1 (1–3)	0.606
Q51	Overall opinion EM practicals ^(c)^	9 (8–10)	9 (7.5–10)	9 (8–10)	9 (7–9)	0.398
Q52	Overall opinion EM lectures ^(c)^	9 (7–10)	8 (6–10)	9 (7–10)	8 (6–9)	0.559
Q53	Video 360 familiar ^(b)^	3 (2–4)	3 (1–4)	3 (2–4)	3 (1–3)	0.089
Q54	Video 360 useful ^(b)^	4 (3–5)	4 (3–5)	4 (3–5)	4 (3–4)	0.492

^(a)^ median (IQR) for the totals (sum of the items); Kruskal–Wallis statistical test for significance of observed differences between the three gender groups; ^(b)^ rank scores between 1 and 5; median (IQR); Kruskal–Wallis statistical test for significance of observed differences between the three gender groups; ^(c)^ mark between 1 and 10; median (IQR); Kruskal–Wallis statistical test for significance of observed differences between the three gender groups. Notation: IQR, inter-quartile range.

**Table 3 ijerph-19-15958-t003:** Associations concerning the BMI and biostatistics courses: two-by-two correlation matrix between the measured residual knowledge, self-assessed level of knowledge, opinions of the examination approach and usefulness of previous courses and overall marks students gave to each course.

Variable		Residual Knowledge ^#^	Self-Assessed Knowledge	Examination Approach	Usefulness of Data Science	BMI Mark	Biostats Mark
Residual knowledge ^#^	R	1.000	0.107	0.029	−0.071	−0.036	0.013
	p	.	0.246	0.753	0.444	0.695	0.889
	N	120	120	120	120	120	120
Self-assessed knowledge	R	0.107	1.000	**0** **.699 ****	**0** **.529 ****	**0** **.589 ****	**0** **.691 ****
	p	0.246	.	<0.001	<0.001	<0.001	<0.001
	N	120	121	121	121	121	121
Examination approach	R	0.029	0.699 **	1.000	**0** **.574 ****	**0** **.572 ****	**0** **.634 ****
	p	−0.753	<0.001	.	<0.001	<0.001	<0.001
	N	120	121	121	121	121	121
Usefulness of data science	R	−0.071	0.529 **	0.574 **	1.000	**0** **.591 ****	**0** **.581 ****
	p	0.444	<0.001	<0.001	.	<0.001	<0.001
	N	120	121	121	121	121	121
BMI mark	R	−0.036	0.589 **	0.572 **	0.591 **	1.000	**0** **.869 ****
	p	0.695	<0.001	<0.001	<0.001	.	<0.001
	N	120	121	121	121	121	121
Biostats mark	R	0.013	0.691 **	0.634 **	0.581 **	0.869 **	1.000
	p	0.889	<0.001	<0.001	<0.001	<0.001	.
	N	120	121	121	121	121	121

Significant R values over 0.5 are in bold. Notations: BMI, biomedical informatics; N, number of paired values in the correlation analysis; p, *p*-value for statistical significance; R, Spearman coefficient of correlation; #, one missing value for a male respondent; **, statistical significance, *p* < 0.01.

**Table 4 ijerph-19-15958-t004:** Associations between the EM lectures and practical activities and the two data science courses: two-by-two correlation matrix between the overall marks students gave to each course or activity.

Variable		EM Practicals Mark	EM Lectures Mark	BMI Mark	Biostats Mark
EM practical mark	R	1.000	**0** **.730 ****	0.477 **	0.387 **
	p	.	<0.001	<0.001	<0.001
	N	121	121	121	121
EM lectures mark	R	**0** **.730 ****	1.000	0.362 **	0.256 **
	p	<0.001	.	<0.001	0.005
	N	121	121	121	121
BMI mark	R	0.477 **	0.362 **	1.000	**0** **.869 ****
	p	<0.001	<0.001	.	<0.001
	N	121	121	121	121
Biostats mark	R	0.387 **	0.256 **	**0** **.869 ****	1.000
	p	<0.001	0.005	<0.001	.
	N	121	121	121	121

Significant R values over 0.5 are in bold. Notations: BMI, biomedical informatics; EM, emergency medicine; N, number of paired values in the correlation analysis; p, *p*-value for statistical significance; R, Spearman coefficient of correlation; **, statistical significance, *p* < 0.01.

## Data Availability

The curated data file is openly available at https://doi.org/10.6084/m9.figshare.20358966.v1.

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
