# Peer review of "Beyond the Digital Competencies of Medical Students: Concerns over Integrating Data Science Basics into the Medical Curriculum"

_ijerph, 2022, doi:10.3390/ijerph192315958_

Round 1
Reviewer 1 Report
The article responds to current challenges that should also be taken into account by the management of higher education.
The introduction of the article offers an adequate introduction to the issue.
The methodology and analysis of the obtained data is described in detail, which provides the reader with enough information to assess the correctness of the research work.
In the Discussion section, I expect a more intensive "communication" of the opinions of various currents of opinion regarding the examined issue.
What recommendations could the authors propose for educational practice based on the obtained research data?
A valuable attribute of the presented article could be the controversy to what extent the subjective opinions of students should decide on the content and form of study programs. After all, universities are responsible for the quality of the educational process. If we were to absolutize the opinions of students, it could happen that in some cases the school would have only minimal requirements.
The article's biggest shortcoming is the small research sample.
Author Response
Please see the attached file.
Thank you, DL

Reviewer 2 Report
Excellent study and presentation of results and conclusions. I would have liked to see more courses and students in order to make the results and conclusions much more credible.
In the conclusions, there is mention of active learning incorporation. I would also add self-directed learning and forced learning.
Author Response

(The authors gave the same response as above.)

Reviewer 3 Report
Although it is a topic of interest for the medical higher education, the paper lacks the clear presentation of the research questions or the hypothesis of the study. Therefore, despite the fascinating presentation of the status quo of the topic discussed, the readers may not understand the use of knowing them.
The instruments are well presented in the supplementary material, but some details should be presented in the paper, too.
Author Response

(The authors gave the same response as above.)

Reviewer 4 Report
Review Report - Beyond the Digital Competencies of Medical Students: Concerns over Integrating the Data Science Basics into the Medical Curriculum
The paper presents the results of an anonymous cross-sectional survey conducted among students in regard to the courses they previously attended, in contrast with the ongoing course of emergency medicine (EM) during the first semester of the academic year 2020 – 2021, when all activities went on-line. The students appreciated the EM course more than the BMI and Biostatistics, due to its value to the medical profession. There is no correlation between measured and self-assessed knowledge of data science, the latter being fairly and significantly correlated with the perceived usefulness of the courses. The study shows that medical students need to see the explicit connection between interdisciplinary or complementary courses and the medical profession and the courses on information technology and data science would better suit in a distributed approach across the medical curriculum.
The study is well and clearly designed, the hypotheses are also clearly formulated and checked, the methodology is accurate and correct, being presented with all the necessary details concerning the procedure, as well as the statistical analysis and the study’s results are also clearly and accurately described. The authors made a comparative study on genders, which is enough taking in consideration that the students belong to the same age group (and the same year of study) – it is interesting for the authors to extend their study (in future papers) in other universities as well, from Romania or abroad, in order to get a more comprehensive image about the investigated topics. It is interesting that the authors detected a small sample of students (9) which preferred to not declare their gender, and these subjects were included into a separated sample to be analyzed – this is a novelty, since most scientific studies make comparisons between males and females.
The paper’s topic is hot and relevant, taking in consideration that in the last two years all education was switched online, with certain advantages and difficulties to be further investigated.
The manuscript is clear and well-structured, the experimental design is certainly appropriate to test the hypothesis and the methodology provides all the necessary details to reproduce the results if necessary. The statistical analysis is well and appropriate conducted, the data are interpreted appropriately and consistently, being easy to understand; the tables provided are relevant and present the results in a proper, accurate and clear manner.
The introduction section is concise and outlines clearly the topic’s state-of-the-art, not only in Romania, but from a general point of view; the authors also present a competent justification for their study and their research objectives, which are also clearly described.
As I said before, the methodology provides all the necessary details and the results section described clearly and also accurate the relevant outcomes.
In which concerns the Discussions section, the authors provide their own pertinent comments but also they compare their results with the results obtained in similar studies worldwide; their findings confirm other researches related to education during the COVID-19 pandemic and students' struggle to adapt, as well as issues regarding the basis of present medical education. The study’s main limitation, in authors opinion, consists in the cross-sectional design of the survey and the limited number of questions in each of its sections, which led to a low reliability of the residual knowledge assessment.
The Conclusions are highly consistent with the evidence and the arguments presented, they are specified in a concise and clear manner and claim the need for providing adequate medical background and rationale behind all complementary courses in the medical curriculum, and behind the biomedical data science in particular. The results bring evidences to support the broadening of educational approaches and the incorporation of contextual learning and active learning into the interdisciplinary courses of the medical curriculum – this is a hot topic for the modern medical learning.
The cited references are recent (most of them being within the last 5 years), extensive and relevant for the discussed topic, with only 2 self-citations, which is acceptable.
The ethics statement and the data availability statement are adequate.
The manuscript fits the journal scope and is interesting for the readership of the journal – it presents local results within a University from Romania, but these findings can be used for comparison in further studies – therefore the authors work can certainly advance the current knowledge in the area; it is also important to point out that the authors address an important long-standing question for the modern learning, which, in my personal opinion, passes nowadays through a paradigm shift toward a virtual environment.
The English language is appropriate and understandable.
Author Response

(The authors gave the same response as above.)

Reviewer 5 Report
Dear Authors,
The study title is beyond the digital competencies of medical students: concerns over integrating the data science basics into the medical curriculum,
There is some advice and suggestions the authors clarify below,
1. Introduction section, the study-related framework is weak, what’s the study’s Dependent variable, Independent variable (DV/IV) are unclear, for more contributions, suggests authors add related references to support each DV/IV “(a) to assess the residual knowledge of BMI and biostatistics; (b) to collect students' opinion and attitude towards the two previous courses of data science as opposed to the EM course; (c) to analyze the possible pandemic-related confounders or determinants of individual perceptions in regard to the quality of the above-mentioned courses, such as students’ overall satisfaction with life, perception of their work effectiveness and on-line professional activity, level of depression, health, and support from the University.”
In addition, why authors want analysis residual knowledge of BMI and biostatistics, satisfaction with life, work effectiveness and online professional activity, level of depression, health, and support, etc. the section lack strengthen reasons.
2. Materials and Methods section, please address the study’s sample size.
3. Study Design should add described related Study flow diagram content, such as medical curriculum via intervention detail process, and how to recruit participants, should address more.
4. Authors stated that “Depression was measured with the nine-item Patient Health Questionnaire 117 (PHQ-9),” but the study’s Participants are focused on students, thus the Questionnaire should consider it’s probably unsuitable.
5. Page 3 “…Residual knowledge of BMI and biostatistics was gauged with 14 multiple choices questions …. Students' opinions and feedback were collected on five-point Likert-type scales.….” Please add measure tool-related references support.
6. “…Residual knowledge of BMI and biostatistics was gauged with 14 multiple choices questions, and the answers were manually coded by two independent reviewers. Students' opinions and feedback were collected on five-point Likert-type scales. Overall opinion of each course was recorded as integer marks between 1 and 10….” Please add the above reviewer’s reliability and validity.
7. Due to the study presented, “The EM course during which the 157 questionnaire was distributed enrolled 79 students,” thus all tables should clarify the real participants, thus results section is incorrect.
8. Due to unclear DV/IV, Study Design, and real participants, thus Discussion and Conclusions section should be reworded.
Thank you for your efforts.
Author Response

(The authors gave the same response as above.)

Round 2
Reviewer 3 Report
Congratulations!
Author Response
Dear Reviewer,
Thank you very much.
On behalf of the Authors,
Faithfully yours, Diana Lungeanu
Reviewer 5 Report
DEAR Authors,
The authors do not all resolve my question, Thus, I suggest the authors should clarify some of my initial issues, such as the revised articles stating that "This investigation was an exploratory analysis," but lack clear DV/IV, and sample sizes. Thank you.
Author Response
Dear Reviewer,
We have to disagree with you, in the sense that we answered all your comments and observations, point by point, in the response we provided to you in round 1 of the reviewing process:
Answer 1 to: the study-related framework + Dependent variable, Independent variable (DV/IV) are unclear
Answer 2 to: the study’s sample size.
Answer 8 to: Due to unclear DV/IV, Study Design, and real participants, thus Discussion and Conclusions section should be reworded
Although you keep checking the option "Extensive editing of English language and style required", we can assure you that the manuscript underwent professional English editing (MDPI Certificate 54442).
It is impossible to us to forcibly cast our manuscript into the Procrustean bed of an interventional design, which does not fit our exploratory investigation and is an unsuitable pattern for our research.
To better clarify what an exploratory analysis is, please consider reading the papers:
John W. Tukey (1980). We need both exploratory and confirmatory, The American Statistician, 34:1, 23-25. DOI: 10.1080/00031305.1980.10482706.
Ronald L. Wasserstein & Nicole A. Lazar (2016) The ASA statement on p-values: context, process, and purpose, The American Statistician, 70:2, 129-133, DOI: 10.1080/00031305.2016.1154108.
Ronald L. Wasserstein, Allen L. Schirm & Nicole A. Lazar (2019). Moving to a world beyond “p < 0.05”, The American Statistician, 73:sup1, 1-19, DOI: 10.1080/00031305.2019.1583913.
The following book might also be useful in this regard:
David C. Hoaglin, Frederick Mosteller, John W. Tukey (1991). Fundamentals of Exploratory Analysis of Variance. New York: John Wiley & Sons. ISBN 0-471-52735-1.
Even the Wikipedia article https://en.wikipedia.org/wiki/Exploratory_data_analysis might be of interest to shed some light into the concept.
We hope that the issues you keep raising in disregard to our answers are rooted in an honest misunderstanding of our investigation, and we look forward to any further enquiries you may have.
Thank you for your efforts and contribution to improving the communication of our endeavor.
On behalf of the Authors,
Faithfully yours,
Diana Lungeanu, PhD